DATA RELEASE

# The genome of the sapphire damselfish *Chrysiptera cyanea*: a new resource to support further investigation of the evolution of Pomacentrids

Emma Gairin[1,*], Saori Miura[1], Hiroki Takamiyagi[1], Marcela Herrera[1,†] and Vincent Laudet[1,2,3,†]

1 Marine Eco-Evo-Devo Unit, Okinawa Institute of Science and Technology, 904-0495, Onna-son, Okinawa, Japan

2 Marine Research Station, Institute of Cellular and Organismic Biology, Academia Sinica, 23-10, Dah-Uen Rd, Jiau Shi, I-Lan 262, Taiwan

3 CNRS IRL 2028 "Eco-Evo-Devo of Coral Reef Fish Life Cycle" (EARLY), France

## ABSTRACT

The number of high-quality genomes is rapidly increasing across taxa. However, it remains limited for coral reef fish of the Pomacentrid family, with most research focused on anemonefish. Here, we present the first assembly for a Pomacentrid of the genus *Chrysiptera*. Using PacBio long-read sequencing with 94.5× coverage, the genome of the Sapphire Devil, *Chrysiptera cyanea*, was assembled and annotated. The final assembly comprises 896 Mb pairs across 91 contigs, with a BUSCO completeness of 97.6%, and 28,173 genes. Comparative analyses with chromosome-scale assemblies of related species identified contig-chromosome correspondences. This genome will be useful as a comparison to study specific adaptations linked to the symbiotic life of closely related anemonefish. Furthermore, *C. cyanea* is found in most tropical coastal areas of the Indo-West Pacific and could become a model for environmental monitoring. This work will expand coral reef research efforts, highlighting the power of long-read assemblies to retrieve high quality genomes.

**Subjects** Animal and Plant Sciences, Marine Biology, Evolutionary Biology

**Submitted:** 03 October 2024

\* Corresponding author. E-mail: emma.gairin@hotmail.fr

† Co senior authors.

Preprint submitted at https://doi.org/10.1101/2024.11.06.622371

## INTRODUCTION

Damselfish (Pomacentridae family) are highly abundant and common demersal reef fish throughout temperate, subtropical, and tropical waters. Many damselfish species, such as anemonefish, are prominent in the aquariology trade worldwide. They are also commonly studied to answer a number of scientific questions, notably at the ecological, behavioural, and developmental levels. As a key resource to deepen scientific analyses, genomes have been generated for fifteen species of Pomacentrids, including ten *Amphiprion* species [1–7]. Chromosome-level genomes are currently available for five species: four Pomacentrinae (*Acanthochromis polyacanthus* [8], *Amphiprion ocellaris* [4], *Amphiprion percula* [1], and *Amphiprion clarkii* [9]) and one Chrominae, *Dascyllus trimaculatus* [3]. To expand the genomic resources of Pomacentrids and reduce the bias for *Amphiprion*, we present here the first genome for the Cheiloprionini *Chrysiptera cyanea*, a member of the Pomacentrinae subfamily.

The Sapphire Devil *C. cyanea* [10] is a strikingly blue damselfish (Figure 1A) that can be encountered in shallow (0–10 m) Indo-West Pacific tropical coral reefs [11] (Figure 1C). *C. cyanea* is territorial, reef-associated, and non-migratory. It is most generally found around rubble and corals in subtidal reef flats and tide pools (Figure 1B). *C. cyanea* has an omnivorous diet, commonly consisting of plankton, algae, and small benthic crustaceans. The size of females typically ranges between 38 and 54 mm total length (TL), while males measure 49–73 mm TL [12]. *C. cyanea* live in small to large schools that typically consist of one or a few males and several females [13, 14] (Figure 1B). The reproductive season of *C. cyanea* in Okinawa lasts from April to August [15, 16], and large densities of juvenile fish can be found in clusters along the coastline during the reproduction season in the presence or absence of adult conspecifics.

The life cycle of *C. cyanea* is akin to the majority of damselfish species, with a pelagic larval duration of 17–21 days [19], followed by a return to the reef coupled with the metamorphosis from larvae to juveniles. The juveniles then grow and eventually become sexually mature females (Figure 1D). The presence of a sex-specific size distribution and bias in the sex ratio of *C. cyanea* on the reef is indicative of protogynous hermaphroditism, with females turning into males [14, 20].

In Okinawa, females have transparent caudal fins and males have opaque blue caudal fins (Figure 1A). This phenotype is different from that of other locations in the Indo-Pacific, where males exhibit a yellow to orange tail. Studies of the cytochrome *c* oxidase subunit I (COI) showed a divergence between *C. cyanea* from Indonesian/Australian, Philippines, and Tonga, suggesting the presence of allopatric divergence with possible new damselfish species (different COI reported in Tonga [21]). *C. cyanea* was initially described in Timor as a blue fish with yellow fins [10]. A synonymous species, *Chrysiptera punctatoperculare* (Fowler, 1946), was described in Aguni, Okinawa. Here, we identified the species living in Okinawa, with blue fins, as *C. cyanea* on the basis of a recent taxonomy guide [22].

This damselfish species is widespread along the coastal zones of Okinawa, making it an easily accessible study model for various behavioural, developmental, ecological, and ecotoxicological questions. To tackle functional and molecular-level questions, in particular those based on gene expression, the availability of a genome is a considerable resource. For instance, in the case of transcriptomic studies, genome availability allows to perform reference-based alignments: such alignments are robust methods to examine RNA sequencing results while minimising issues, such as incorrect assemblies of paralogous sequences (from gene duplication and diversification, leading to families of genes with similar sequences), preventing an underestimation of gene expression levels, as well as avoiding incomplete and fragmented assemblies produced by *de novo* assembly methods [23]. To date, no genomic data is available for any *Chrysiptera* species. Here, we present a highly complete genome obtained from PacBio long-read sequencing, making this study the first to publish a genome for this coral reef fish genus, and adding to the number of genomes available for damselfishes. These genomes can also be useful to improve the analysis of the unique adaptations present in anemonefishes, by comparison with other damselfish [24].

## METHODS

### Fish collection and DNA sequencing

A male *C. cyanea* (total length: 5.6 cm) was collected using SCUBA and hand nets on Sesoko beach in Okinawa, Japan (26.6509 N, 127.8564 E) on September 30th, 2022. The fish was kept

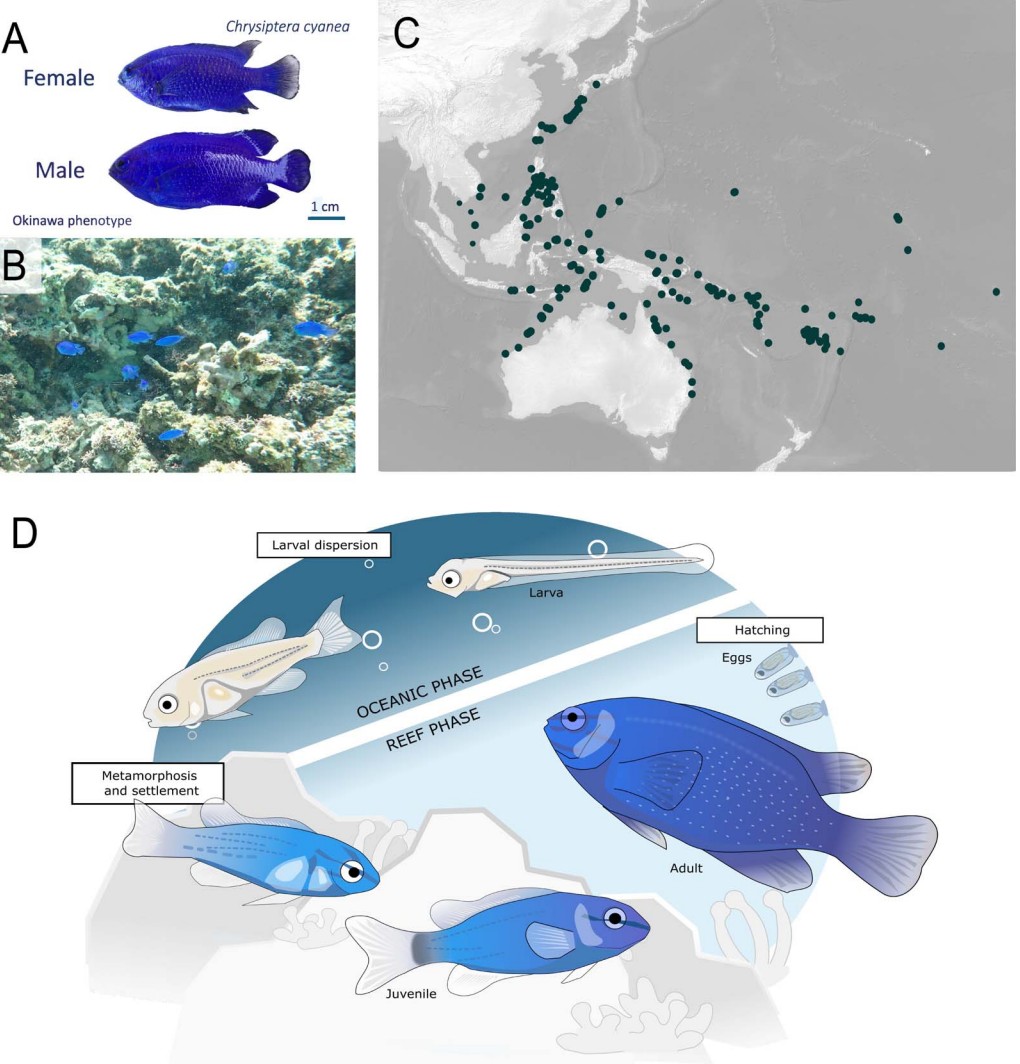

**Figure 1.** (A) Female and male *Chrysiptera cyanea* collected at Sesoko beach, Okinawa, in September 2022. (B) Underwater photography of a school of *C. cyanea* at Sesoko Beach, September 2022. (C) Global georeferenced records of *C. cyanea* (477 records) were extracted from the Wallace module [17] and plotted with QGIS 3.22.9 [18]. (D) Life cycle of *C. cyanea* from hatching to adult stage. Drawing by Stefano Vianello.

under natural conditions in a 270 L flow-through outdoor tank at the OIST Marine Science Station until October 19th, 2022. It was euthanised in a 200 mg/L Tricaine Methanesulfonate (MS222) solution following the guidelines for animal use issued by the Animal Resources Section of OIST Graduate University. Liver tissue destined for genome sequencing was snap-frozen in liquid nitrogen and stored at −80 °C. Eye, gill, liver, and muscle tissue from the same male fish, and brain tissue from another male from the same site, were preserved in RNAlater for transcriptomic sequencing and stored at −30 °C.

Genomic DNA was extracted from the snap-frozen liver using the Monarch HMW DNA Extraction Kit for tissues (New England BioLabs). Library preparation and sequencing were performed by the OIST Sequencing Section. First, the genomic DNA was sheared to 25 kb using the Megaruptor 3 (Diagenode), as longer reads improve the ease of genome assembly.

Library preparation was conducted using the SMRTbell Express Template Prep Kit 3.0 (PacBio, #102-182-700) following the manufacturer's protocol. Library size selection was carried out using the BluePippin system (SageScience, MA USA). After preparation with the Sequel Binding Kit 3.2 to bind the polymerase and sequencing primers, sequencing was performed on two SMRT cells with Circular Consensus Sequencing runs using PacBio Sequel IIe HiFi (Pacific Biosciences, CA, USA).

RNA was extracted from the eye, gill, liver, muscle, and brain tissue using the Maxwell® RSC simplyRNA tissue kits and an automated Maxwell® RSC instrument, following manufacturer recommendations (Promega, Cat. No. AS2000, Wisconsin, USA). The quality and concentration of RNA were assessed with an Invitrogen Qubit Flex benchtop fluorometer and an Agilent 4200 TapeStation. Library preparation was performed by the OIST Sequencing section using the NEBNext Ultra II Directional RNA library prep kit for Illumina (New England BioLabs, USA). Sequencing was performed using an Illumina NovaSeq6000 platform at OIST.

### Sequencing data processing and genome assembly

The complete list of software, version, and parameters is provided in Table 1. Quality control was performed with FastQC 0.11.9 (RRID:SCR_014583) on the raw reads from each SMRT cell. The genome size was estimated based on a k-mer approach using Jellyfish 2.2.7 (RRID:SCR_005491) and GenomeScope 2.0 (RRID:SCR_017014). The DNA sequences were assembled with the Improved Phased Assembler 1.3.1 (IPA) [25] without phasing – i.e., without separating the parental alleles into haplotypes, as this option yielded the best assembly results out of different IPA and Flye 2.9.1 (RRID:SCR_017016) [26] assemblies based on Quast 5.2.0 (RRID:SCR_001228) statistics [27], BUSCO (Benchmarking Universal Single-Copy Orthologs) (RRID:SCR_015008) scores (actinopterygii_odb10, BUSCO 4.1.2) [28], and similarity to the k-mer estimation of the genome size. Merqury 1.3 (RRID:SCR_022964) was used to estimate the genome completeness and error rate. Purge Haplotigs 1.1.3 was used to improve contiguity by removing allelic contigs from the non-phased IPA assembly [29]. The repeats were annotated with RepeatModeler 2.0.3 (RRID:SCR_015027) with the parameter -LTRStruct [30]. RepeatMasker 4.1.1 (RRID:SCR_012954) [31] was run to identify repetitive elements in the RepeatModeler output and the vertebrate library of Dfam. Repetitive elements were then softmasked with BEDTools 2.30.0 (RRID:SCR_006646) [32].

### Contig scaffolding on clownfish reference genomes

Using MUMmer 3.23 (RRID:SCR_018171) [33], the contigs of the *C. cyanea* assembly were mapped to the chromosome-scale genome assemblies for *A. clarkii*, *A. ocellaris*, *A. percula*, *Ac. polyacanthus*, and *D. trimaculatus*. Dot plots were generated using ggplot2 (RRID:SCR_014601) [34] after data filtering to remove any alignments shorter than 10,000 bases [35].

### Transcriptome sequencing data processing

Quality control of the RNA sequencing data from the eye, gill, liver, muscle, and brain tissue was performed with FastQC 0.11.9 [36]. Following this quality check, the sequences were processed by trimming the adapters used by Illumina on each transcript as well as dropping low quality sections with Trimmomatic 0.39 (RRID:SCR_011848) [37]. The transcriptomic reads were mapped to the contig sequences with HISAT2 2.2.1

**Table 1.** Software and parameters used for the genome assembly and annotation of *Chrysiptera cyanea*.

| Purpose | | Software and version | Parameters |
|---|---|---|---|
| Raw reads quality check | | FastQC 0.11.9 | |
| Estimate genome size | | Jellyfish 2.2.7 | $k$ = 21, $s$ = 100 M |
| Reference-free characterization | | GenomeScope 2.0 | |
| *De novo* assembly | | Improved Phased Assembler 1.3.1 | Phase and no phase |
| | | Flye 2.9.1 | With and without haplotypes, with and without scaffolding |
| Haplotigs removal | | Purge Haplotigs 1.1.3 | low = 10, $m$ = 75, $h$ = 195 |
| Repeat modeling and masking | | RepeatModeler 2.0.3 RepeatMasker 4.1.4 | LTRStruct |
| Genome annotation | | Hisat 2.2.1 Braker 2.1.6 | Protein evidence from UniProtKB/Swiss-Prot, ten related genomes |
| | | Diamond 2.0.14 InterProScan 5.60 | Swissprot Pfam |
| Functional annotation | | NCBI Blast 2.10.0 | |
| File processing | | Bedtools 2.29.2 Samtools 1.12 | |
| Mapping | | Minimap 0.2 | |
| Assembly statistics | General metrics Completeness Quality and error rate | Quast 5.2.0 BUSCO 4.1.2 Merqury 1.3 | actinopterygii_odb10 |
| Check for contamination | | Blobtools 1.1 | |

(RRID:SCR_015530) [38], then converted to BAM format from SAM format with SAMtools 1.12 (RRID:SCR_002105) [39, 40]. Lastly, the number of transcripts per gene was quantified using Kallisto 0.46.2 (RRID:SCR_016582) [41].

## Prediction of gene models

The position of protein-coding gene structures in the soft-masked assembly was predicted by BRAKER 2.1.6 (RRID:SCR_018964) [39, 42–53] using the transcriptome of the eye, gill, liver, and muscle of the fish used for the genome. The position of the protein-coding gene structures was also predicted using protein evidence from UniProtKB/Swiss-Prot [54] as well as selected fish proteomes from the NCBI database (*A. ocellaris*: 48,668 sequences; *A. clarkii*: 25,025; *Danio rerio*: 88,631; *Ac. polyacanthus*: 36,648; *Oreochromis niloticus*: 63,760; *Oryzias latipes*: 47,623; *Poecilia reticulata*: 45,692; *Stegastes partitus*: 31,760; *Takifugu rubripes*: 49,529; and *Salmo salar*: 112,302). We selected genes with evidence from the transcriptome or protein hints, and with homology to the Swiss-Prot protein database and Pfam domains identified with Diamond (RRID:SCR_016071) [45] and InterProScan (RRID:SCR_005829) [55]. NCBI BLAST (RRID:SCR_004870f) 2.10.0 [56] was used to perform the functional annotation of the final gene models.

The output was assessed using Quast 5.2.0 [27] as well as by calculating the number of BUSCO genes present in the assembly [28].

## Gene expression analysis

The tissue-specify index ($\tau$) was calculated for each gene using the R package tispec v0.99 [57]. The relationship between $\tau$ and the transcripts per million (TPM) gene-expression values was visualized on a 2D histogram with ggplot2 [34]. An upset plot from UpSetR v1.4.0 [58] was used to visualise the TPM values per tissue (brain, eye, gill, liver, and muscle).

## RESULTS

### Genome assembly of *C. cyanea*

We assembled the genome of the damselfish *C. cyanea* by sampling one individual from Okinawa and obtaining 3,335,935 PacBio reads. The average read length was 25,387 for a total of 84,688,690,513 sequenced bases. FastQC 0.11.9 did not detect low-quality reads (Figure 2).

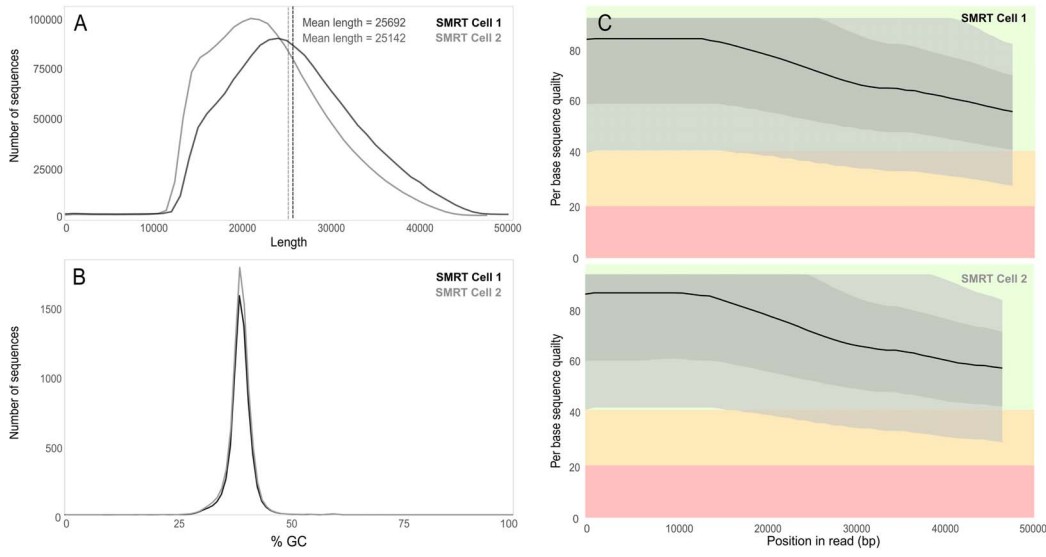

**Figure 2.** (A) Distribution of the raw read length from the two SMRT cells. (B) Distribution of the GC content across the sraw reads from the two SMRT cells. (C) Per base sequence quality of the raw reads from the two SMRT cells; data obtained using FastQC 0.11.9.

Multiple *de novo* assembly options were tested (Table 2) using Flye and IPA. We chose the IPA no-phasing primary assembly as it yielded the highest N50 (30.6 Mb), the lowest number of contigs (103), and a genome length of 899.7 Mb. From the raw reads, the genome size had been estimated with a k-mer approach to be 753–754 Mb (using Jellyfish and GenomeScope with k-mer sizes 21 and 31; Figure 3, Supplementary Table 1 in Figshare [60]). This estimate was 16% lower than the assembly; the discrepancy might be due to the high repeat content of teleost fish genomes, leading to an underestimation of the genome size using a k-mer approach. Similarly, the estimated size of the genome using the k-mer approach was lower than the final assembly for *A. ocellaris* [4], *A. clarkii* [9], and *D. trimaculatus* [3]. Here, the assembly size was close to the reported lengths of other damselfish genomes. For instance, the genome size of *A. ocellaris* is 861.4 Mb [4], of *A. clarkii* is 843.6 Mb [9], of *A. percula* is 908.8 Mb [1], and of *D. trimaculatus* is 910.8 Mb [3].

The *de novo* assembly was curated using Purge Haplotigs (removing one contig when syntenic pairs of contigs were detected) [29]. 12 contigs were removed to obtain a curated assembly with 91 contigs, with a total length of 896.5 Mb. The GC content of the genome was 38.73%, and the mean base-level coverage was 94.5× (Table 2). We retrieved 97.6% of BUSCO genes in the final genome assembly, with 96.9% single copy and 0.7% duplicated. We found 0.6% of BUSCO genes to be fragmented and 1.8% missing. The completeness of the genome estimated by Merqury was 90.129%, with a mean quality value of 48.9 and an error rate of 0.0000138 (one nucleotide error per 72.6kb).

**Table 2.** Statistics generated with Quast 5.2.0 and BUSCO 4.1.2 about the different primary genome assemblies produced by the Flye 2.9.1 and the IPA 1.3.1, as well as about the final assembly, which was obtained from the non-phased purged assembly (read depth cutoff: low = 10, medium = 70, high = 195) generated with Purge Haplotigs 1.1.3, with soft-masked repeats obtained from RepeatModeler 2.0.3 and RepeatMasker 4.1.4.

| | Flye | | | | IPA | | | | |
|---|---|---|---|---|---|---|---|---|---|
| | With haplotypes | Without haplotypes | With haplotypes, scaffold | Without haplotypes, scaffold | Phase A | Phase P | No phase A | No phase P | No phase P purged |
| **Number of contigs** | **3,392** | **1,705** | **2,788** | **1,677** | **1,506** | **126** | **5,516** | **103** | **91** |
| Largest contig | 16,926,528 | 31,064,569 | 24,238,470 | 32,994,236 | 31,083,763 | 70,778,723 | 5,766,530 | 56,011,215 | 56,011,215 |
| **Total length** | **895,607,061** | **907,362,611** | **902,966,492** | **907,614,972** | **559,243,152** | **1,252,663,990** | **609,931,055** | **899,747,371** | **896,528,091** |
| GC (%) | 38.72 | 38.72 | 38.73 | 38.72 | 38.08 | 38.92 | 38.4 | 38.74 | 38.73 |
| **N50** | **3,366,676** | **6,245,761** | **2,428,049** | **7,852,259** | **8,678,642** | **22,721,003** | **153,437** | **30,591,387** | **30,591,387** |
| N90 | 256,067 | 687,672 | 268,683 | 710,351 | 89,761 | 5,200,624 | 47,219 | 4,184,431 | 4,251,551 |
| L50 | 66 | 34 | 86 | 32 | 19 | 20 | 708 | 12 | 12 |
| L90 | 476 | 198 | 524 | 179 | 176 | 61 | 3,718 | 39 | 38 |
| Number of N's per 100 kb | 0 | 0 | 0 | 0.34 | 0.02 | 0.02 | 0.01 | 0.03 | 0.03 |
| Number of contigs (≥1,000 bp) | 3,295 | 1,701 | 2,751 | 1,673 | 1,506 | 126 | 5,516 | 103 | 91 |
| Total length (≥1,000 bp) | 895,534,291 | 907,359,480 | 902,937,845 | 907,612,138 | 559,243,152 | 1,252,663,990 | 609,931,055 | 899,747,371 | 896,528,091 |
| Number of contigs (≥5,000 bp) | 2,631 | 1,591 | 2,435 | 1,562 | 1,505 | 126 | 5,509 | 103 | 91 |
| Total length (≥5,000 bp) | 893,580,031 | 907,017,243 | 901,992,706 | 9072,72,034 | 559,241,105 | 1,252,663,990 | 609,902,971 | 899,747,371 | 896,528,091 |
| Number of contigs (≥10,000 bp) | 1,909 | 1,396 | 2,110 | 1,359 | 1,495 | 126 | 5,455 | 103 | 91 |
| Total length (≥10,000 bp) | 888,503,131 | 905,565,959 | 899,619,340 | 905,751,218 | 559,155,404 | 1,252,663,990 | 609,468,326 | 899,747,371 | 896,528,091 |
| Number of contigs (≥25,000 bp) | 1,256 | 875 | 1,418 | 842 | 1,326 | 126 | 5,087 | 103 | 91 |
| Total length (≥25,000 bp) | 878,176,247 | 897,047,373 | 888,532,498 | 897,267,975 | 556,013,220 | 1,252,663,990 | 602,874,429 | 899,747,371 | 896,528,091 |
| Number of contigs (≥50,000 bp) | 978 | 572 | 1,040 | 542 | 539 | 126 | 3,463 | 103 | 91 |
| Total length (≥50,000 bp) | 868,491,235 | 886,407,758 | 875,155,475 | 886,689,354 | 525,520,384 | 1,252,663,990 | 536,558,481 | 899,747,371 | 8,965,28,091 |
| **BUSCO Complete (%)** | **97.4** | **97.7** | **97.3** | **97.6** | **52.8** | **97.7** | **62.0** | **97.5** | **97.6** |
| BUSCO Single copy (%) | 96.4 | 96.3 | 95.8 | 96.2 | 51.2 | 53.6 | 60.6 | 96.5 | 96.9 |
| BUSCO Duplicated (%) | 1.0 | 1.4 | 1.5 | 1.4 | 1.6 | 44.1 | 1.4 | 1 | 0.7 |
| BUSCO Fragmented (%) | 0.9 | 0.8 | 1.2 | 0.7 | 0.5 | 0.4 | 3.9 | 0.7 | 0.6 |
| BUSCO Missing (%) | 1.7 | 1.5 | 1.5 | 1.7 | 46.7 | 1.9 | 34.1 | 1.8 | 1.8 |

A total of 324,222,639 bp (36.16% of sequences) were identified as being repeat content using RepeatMasker, with 20% DNA repeat elements, 18% simple repeats (microsatellites), 6% long interspersed nuclear elements, 2% low complexity repeats, 2% long terminal repeats, and under 1% each of short interspersed nuclear elements, rolling circles, tRNA, retroposon, rRNA, snRNA, and Satellite elements. We could not identify 50% of the repetitive elements in the *C. cyanea* genome.

As a reference, a comparison of key statistics of this genome assembly with previously published chromosome-scale genomes of Pomacentrid fish is provided in Table 3. Overall,

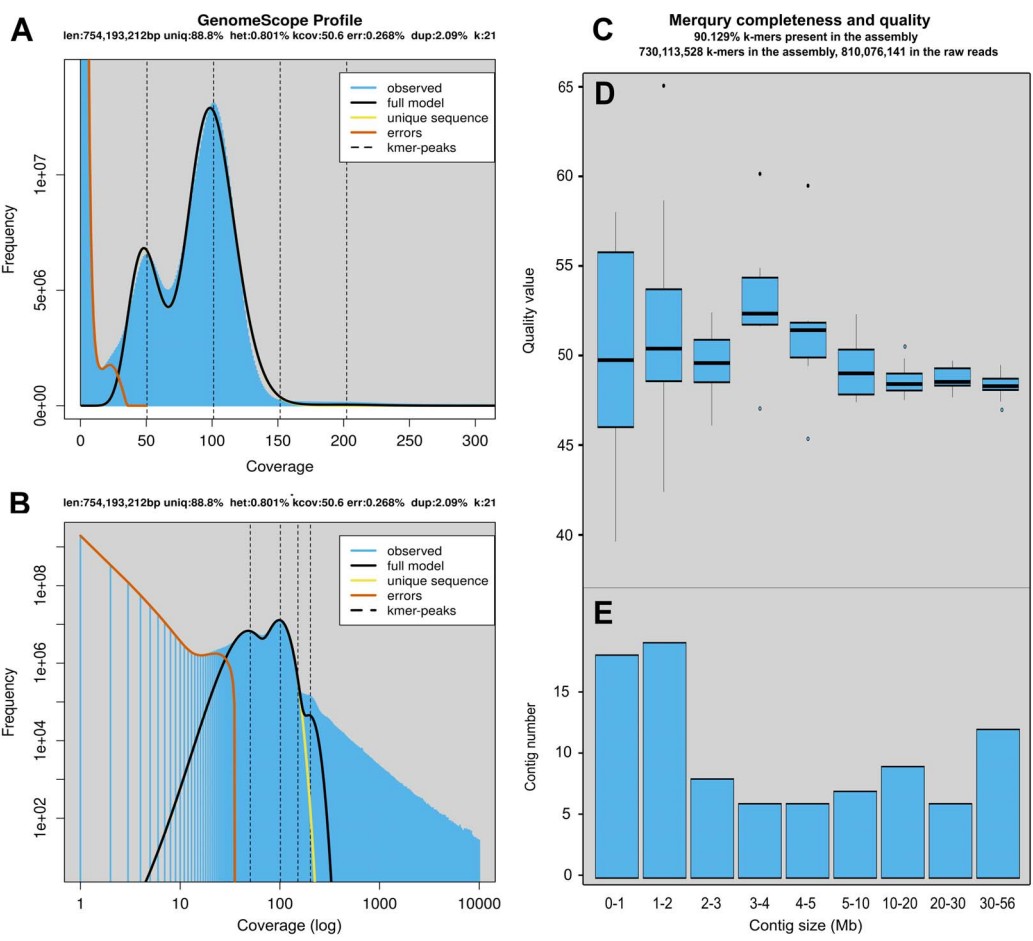

**Figure 3.** (A) Raw coverage plot of the frequency at which k-mers are covered within the raw reads, following k-mer assessment by Jellyfish (k-mer size = 21 bases). Figure generated by GenomeScope. (B) Log-transformed coverage plot of the frequency at which k-mers are covered within the raw reads, following k-mer assessment by Jellyfish (k-mer size = 21 bases). Figure generated by GenomeScope. (C) Key statistics about the final assembly (IPA no phase, purged) from Mercury. (D) Boxplot of the quality value of the contigs based on their size. The boxplot displays the median, 25th and 75th percentiles, with whiskers extending to the maximum and minimum values, within a distance from the box up to 1.5 times the interquartile range from Mercury. (E) Distribution of the contigs in the assembly based on their size.

while our coverage was slightly lower than other genomes, we were able to obtain a much more contiguous *de novo* assembly (91 contigs *vs.* 951 in *Ac. polyacanthus*, and over 1,400 in all other assemblies) by using very long 25 kb DNA fragments for sequencing (rather than the recommended 10–20 kb). Our contig N50 of 30.6 Mb was six to thirty times larger than those of the other assemblies. The GC, repeat contents, and BUSCO completeness scores were similar to those of other genomes (Table 3).

The 91 contigs were mapped against the chromosome-scale genomes of *A. clarkii, A. ocellaris, A. percula, D. trimaculatus*, and *Ac. polyacanthus* using MUMmer. After filtering to remove aligned segments shorter than 10,000 bases, over 40% of the bases of the assembly of *C. cyanea* were matched with the reference genomes of *A. clarkii, A. ocellaris, A. percula,* and *Ac. polyacanthus*. However, only 20.3% were retrieved in the alignment with *D. trimaculatus* (Supplementary Table 4 in Figshare [60]); the latter genome was reported to have undergone rearrangements [3].

**Table 3.** Comparison of the genome assembly strategy and output for *C. cyanea* with other chromosome-scale genome assemblies of Pomacentrid fish.

| | *Amphiprion ocellaris* (false clownfish) | *Amphiprion clarkii* (yellow tail clownfish) | *Amphiprion percula* (orange clownfish) | *Acanthochromis polyacanthus* (spiny chromis) | *Dascyllus trimaculatus* (threespot damselfish) | *Chrysiptera cyanea* (blue damselfish) |
|---|---|---|---|---|---|---|
| Sequencing strategy | PacBio Sequel II, HiC | PacBio Sequel II, HiC | PacBio RSII, HiC | PacBio RSII, HiC | HiSeq4000, 2 × 150 PE | PacBio Sequel II |
| Mean base-level coverage | 103.9× | 250.4× | 121× | 131× | 103× | 94.5× |
| *De novo* assembly method | Falcon | Flye with haplotypes, Purge Haplotigs | Falcon | Falcon | MaSuRCA | IPA non-phased primary assembly, purge haplotigs |
| Assembly size | 856.6 Mb | 843.6 Mb | 908.8 Mb | 956.8 Mb | 910.7 Mb | 896.6 Mb |
| Number of chromosomes | 24 | 24 | 24 | 24 | 24 | NA |
| Number of scaffolds including chromosomes | 353 | 192 | 365 | 81 | 2156 | NA |
| Scaffold N50 | 36.9 Mb | 26.7 Mb | 38.4 Mb | 41.7 Mb | 34.9 Mb | NA |
| Number of contigs | 1,551 | 1,840 | 1,414 | 951 | 3,501 | 91 |
| Contig N50 | 0.9 Mb | 1.2 Mb | 1.9 Mb | 5.5 Mb | 1.1 Mb | 30.6 Mb |
| GC content (%) | 39.58 | 39.71 | 39.5 | 40.5 | 41.5 | 38.7 |
| Repeat content (%) | 45 | 44 | 19 | 38 | not reported | 36 |
| BUSCO genome completeness (%) | 97 | 98.7 | 97.1 | 96.7 | 97.9 | 97.6 |
| Complete and single copy (%) | 96.2 | 97.8 | 96.1 | 94.3 | 96.2 | 96.9 |
| Complete and duplicated (%) | 0.8 | 0.9 | 1.0 | 2.4 | 1.7 | 0.7 |
| Fragmented (%) | 0.5 | 0.4 | 0.5 | 0.8 | 0.7 | 0.6 |
| Missing (%) | 2.4 | 0.9 | 2.4 | 2.5 | 1.4 | 1.8 |
| Number of protein-coding genes | 26,797 | 25,050 | 26,597 | 25,468 | not reported | 28,173 |
| BUSCO gene annotation completeness (%) | 96.6 | 97.0 | 96.2 | 94.2 | not reported | 96.6 |
| Complete and single copy (%) | 95.5 | 96.1 | 84.8 | 76.0 | not reported | 96.0 |
| Complete and duplicated (%) | 1.1 | 0.9 | 11.4 | 18.2 | not reported | 0.6 |
| Fragmented (%) | 1.0 | 1.1 | 2.0 | 3.2 | not reported | 1.5 |
| Missing (%) | 2.3 | 1.9 | 1.7 | 2.6 | not reported | 1.9 |
| Data release | Ryu *et al.* [4] | Moore *et al.* [59] | Lehman *et al.* [1] | Lehman *et al.* [8] | Roberts *et al.* [3] | This study |

48 contigs were mapped onto the reference genomes (filtering out alignments shorter than 10,000 bases). The correspondences are provided in Table 4. 16, 18, 14, 16, and 0 contigs had over 50% of the bases aligning to the reference genomes of *A. clarkii, A. ocellaris, A. percula, Ac. polyacanthus*, and *D. trimaculatus* (Figure 4A). 46 contigs were aligned with a single chromosome (Figure 4A). 43 contigs did not robustly map to any chromosome of the reference genomes, among which contig-36 was the largest contig that could not be matched (4.6 Mb).

Two *C. cyanea* contigs were mapped to more than one chromosome from the reference assemblies: contigs 1 and 2 (the two largest contigs, of sizes 56.0 Mb and 45.2 Mb, respectively). This result may indicate a chimeric assembly in the analysis of the data or genomic rearrangement. For contig-1, the alignment against the *A. ocellaris* genome highlighted a missing zone between the 19,943,083 and 33,394,537th bases with no alignments longer than 10,000 bases (Figure 4C). Similar results were found for comparisons with the other reference genomes. Lowering the filtering threshold to include alignments between 5,000 and 10,000 bases against *A. ocellaris* retrieved some relatively scrambled alignments in the gap mentioned above (Figure 4D). This could indicate a chimeric assembly of contig-1, which may in fact consist of two chromosomes in *C. cyanea*. However, in a comparison with *A. percula* for alignments longer than 5,000 bases, it was found that the missing zone maps onto *A. percula*'s chromosome 20 (Figure 4E). This may

**Table 4.** Correspondence of the contigs of the genome assembly for *C. cyanea* and the chromosomes of the reference genomes of *Dascyllus trimaculatus, Acanthochromis polyacanthus, Amphiprion clarkii, Amphiprion ocellaris*, and *Amphiprion percula*. A total of 223 chromosome-contig matches were retrieved. Matches with less than 100,000 bases are not indicated in the table (50 matches removed). Scaffolding was performed using the nucmer function of the MUMmer 3.23 package [33], removing alignments shorter than 10,000 bases.

| *Chrysiptera cyanea* contig | *A. clarkii* chromosome | *A. ocellaris* chromosome | *Ac. polyacanthus* chromosome | *A. percula* chromosome | *D. trimaculatus* chromosome |
|---|---|---|---|---|---|
| 1 | 10, 16 | 11, 16 | 14, 15 | 12, 14, 20 | 12, 20 |
| 2 | 7, 18 | 9, 20 | 1, 12 | 7, 18 | 7 |
| 3 | 9 | 7 | 13 | 9 | 9 |
| 4 | 2 | 4 | 3 | 3 | 3 |
| 5 | 1 | 1 | 2 | 1 | 1 |
| 6 | 14 | 15 | 11 | 17 | 17 |
| 7 | 3 | 2 | 4 | 2 | 2 |
| 8 | 8 | 8 | 6 | 8 | 8 |
| 9 | 20 | 18 | 19 | 19 | 19 |
| 10 | 19 | 17 | 18 | 16 | 16 |
| 11 | 22 | 22 | 20 | 22 | 22 |
| 12 | 17 | 19 | 21 | 14 | 14 |
| 13 | 5 | 5 | 5 | 6 | 6 |
| 14 | 11 | 10 | 9 | 10 | 10 |
| 15 | 21 | 21 | 1 | 21 | 21 |
| 16 | 4 | 3 | 8 | 4 | 4 |
| 17 | 6 | 6 | 7 | 5 | 5 |
| 18 | 15 | 14 | 17 | 15 | 15 |
| 19 | 12 | 13 | 10 | 13 | 13 |
| 20 | 18 | 20 | 1 | 18 | 18 |
| 21 | 13 | 12 | 16 | 11 | 11 |
| 22 | 15 | 14 | 17 | 15 | NA |
| 23 | 10 | 11 | 14 | 12 | 12 |
| 24 | 12 | 13 | 10 | 13 | 13 |
| 25 | 23 | 23 | 23 | 23 | 23 |
| 26 | 13 | 12 | 16 | 11 | NA |
| 27 | 4 | 3 | 8 | 4 | NA |
| 29 | 24 | 24 | 22 | 24 | NA |
| 30 | 6 | 6 | 7 | 5 | 5 |
| 31 | 18 | 20 | 1 | 18 | 18 |
| 32 | 13 | 12 | 16 | 11 | NA |
| 41 | 11 | 10 | 9 | 10 | NA |
| 47 | 23 | NA | NA | NA | NA |
| 51 | 23 | 23 | 23 | 23 | NA |
| 54 | 24 | 24 | 22 | 24 | NA |

point towards genomic rearrangement from two or more ancestral chromosomes, with divergence through evolution leading to the low mapping rate between 19 Mb and 33 Mb on contig-1.

## *C. cyanea* gene annotation

The genome was annotated using BRAKER v.2.1.6 based on mRNA and protein evidence, leading to the prediction of 46,873 gene models. These were filtered to keep only the longest isoform of each gene model. We retained genes with mRNA evidence or homology to the Swiss-Prot protein database and Pfam domains. The completeness of the final set of 28,173 genes was assessed using BUSCO, with a final score of 96.6%, including 96.0% single-copy genes and 0.6% duplicated genes. We found that 1.5% of BUSCO genes were fragmented, and 1.9% were missing from the final transcriptome. Of the final set of 28,173 genes, 19,356 (69%) had at least one associated GO term. Lastly, 1,802 genes (6.4% of all genes) were



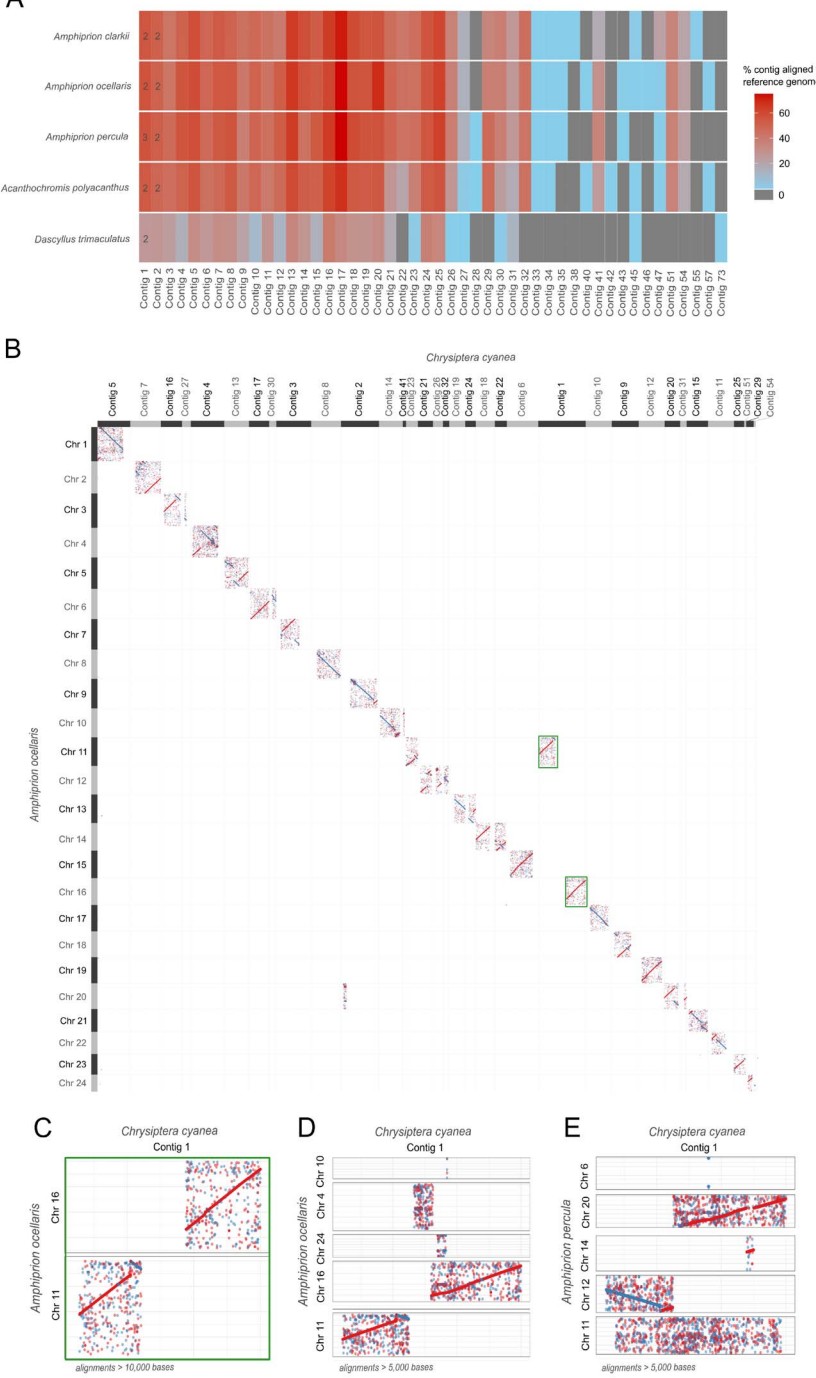

**Figure 4.** (A) Percentage of bases in each contig aligning to the reference genomes. Only contigs aligning to at least one chromosome, with at least 10,000 bases retrieved from the reference genome, are displayed (48 of the total 91 contigs). Based on this threshold, most contigs align to only one reference chromosome. In the case of an alignment to more than one chromosome, the number of retrieved chromosomes is indicated in the corresponding tile of the plot. (B) Dotplot of the alignment of the chromosome-scale *Amphiprion ocellaris* genome with the contigs from *Chrysiptera cyanea* (alignments longer than 10,000 bases). Contig-1 is indicated with a green frame. (C) Dotplot of the alignment of contig-1 from *Chrysiptera cyanea* with chromosomes 11 and 16 of *A. ocellaris* (alignments longer than 10,000 bases). This dotplot is a larger version of the green frame in panel (B). (D) Dotplot of the alignment of contig-1 from *Chrysiptera cyanea* with chromosomes 4, 10, 11, 16, and 24 of *A. ocellaris* (alignments longer than 5,000 bases). (E) Dotplot of the alignment of contig-1 from *Chrysiptera cyanea* with chromosomes 6, 11, 12, 14, and 20 of *Amphiprion percula* (alignments longer than 5,000 bases).

located on contigs that could not be matched to any chromosome from the five reference genomes.

## Tissue specificity of the gene expression

The tissue specificity of the 28,173 identified genes, or the "transcriptomic atlas" of *C. cyanea*, was assessed using the transcriptomes of the eye, gill, liver, and muscle of the fish used for the genome assembly, as well as the brain of another fish collected on the same day and at the same site. In total, 1,639 genes were expressed in all five tissues (with TPM values above 10 in each tissue, Figure 5), similarly to previous reports in anemonefish [4, 59]. The

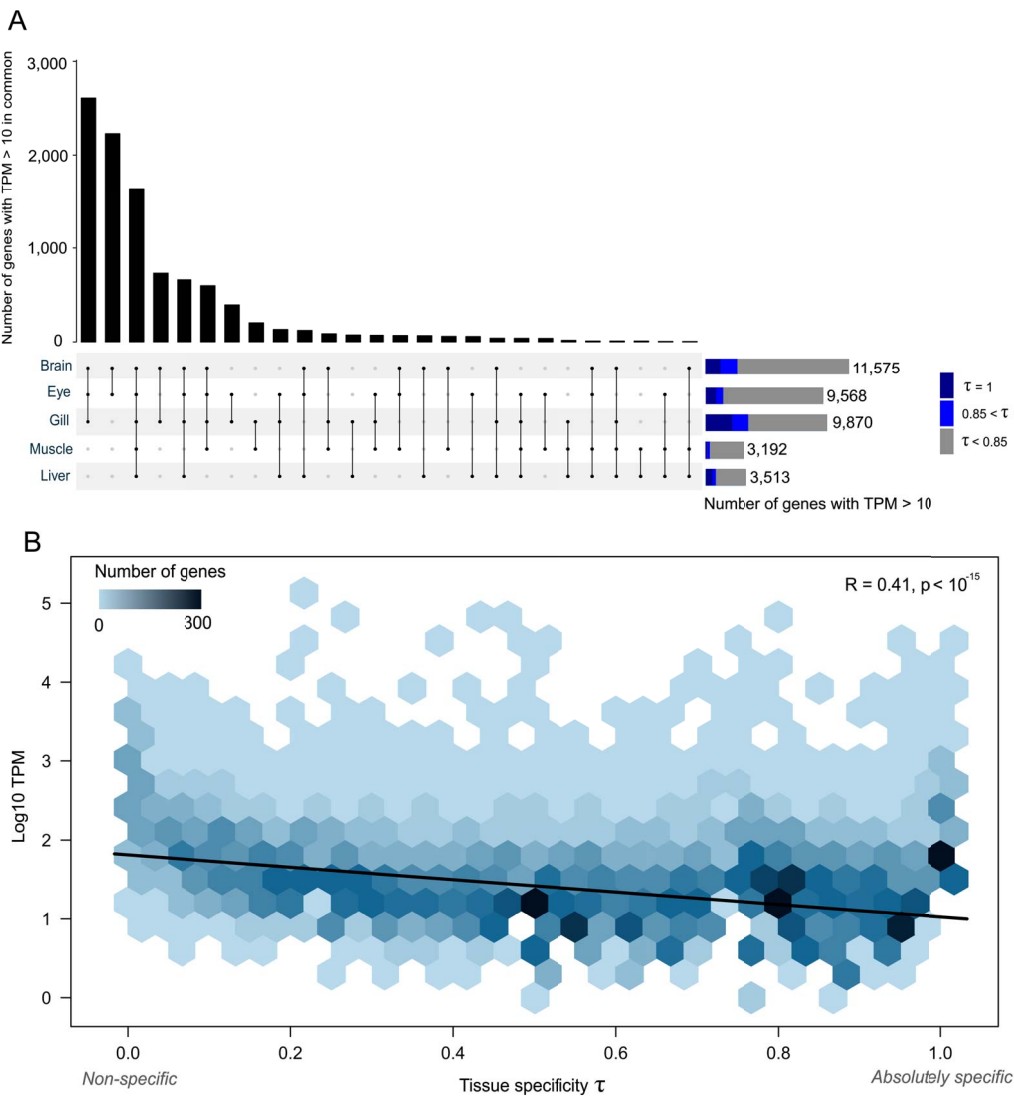

**Figure 5.** (A) Upset plot displaying number of genes expressed (intersection size) in individuals and combinations of different tissues (brain, eye, gill, liver, and muscle) results section. Transcripts per million (TPM) values above 10 were used as a threshold for gene expression. (B) Two-dimensional histogram displaying the relationship between the maximum TPM (log-transformed) and the tissue specificity index ($\tau$) of each gene. The trendline displays the Pearson's correlation between $\tau$ and log-transformed TPM.

tissue specificity index was calculated for each gene (for which near 0 indicates broadly expressed genes, and near 1 specific genes) [61, 62]. A total of 3,090 genes had a value below 0.2, corresponding to housekeeping genes (similar expression levels across most tissues without bias) [61]. This number was relatively similar to those reported for *Amphiprion* species, notably for *A. clarkii* (3,697 genes) [59] and *A. ocellaris* (3,431 genes) [4]. Following log-transformation and normalisation, 7,396 genes with values beyond 0.85, indicative of high tissue specificity, were identified. Of these, 4,258 genes were absolutely specific with a = 1 (expressed in only one type of tissue). This is a relatively high number and may be linked to the low amount of tissue types considered here, compared to other studies of damselfish (five here against, e.g., twelve in [59]). The gill tissue had the highest number of highly specific genes, with 1,170 genes with = 1, followed by the brain tissue (1,094), eye (780), liver (480), and muscle (134). These values were proportional to the total number of genes expressed in each tissue, which was higher in the brain, eye, and gill, whereas it was lower in the liver and muscle, similarly to other damselfish. Lastly, tissue-specific genes tended to have low expression levels (negative correlation of value with gene expression level, $R = 0.41$, $p < 10^{-15}$, Figure 5), as previously reported for *A. clarkii* [59] and *A. ocellaris* [4].

## DISCUSSION

The Sapphire Devil *C. cyanea* is a widely distributed damselfish in the Indo-Pacific area. In Okinawa, it is among the most common species on reef flats [63]. It has been mainly studied to elucidate the roles of various environmental controls on their reproduction and investigate related hormonal processes [12, 15, 16]. To further the potential of biomolecular analyses based on this species, this study generated the first genome of a *Chrysiptera* fish from a male individual collected in Okinawa, Japan. This genome will be of high value for future genetic-based approaches, from population structure to gene expression analyses. Among hot topics in research, the difference between anemonefish and other damselfish is particularly examined. Here, we provide a new high-quality non-anemonefish genome, which will be of relevance to further the depth of such analyses.

Using PacBio HiFi, we assembled a high-quality genome of size 896.5 Mb, within the range of other related species, with a high completeness of 97.6% BUSCO genes. Of particular interest, although Hi-C was not generated for this study, the use of PacBio HiFi long-read technology, the IPA, and Purge Haplotigs allowed us to identify 91 contigs. This is a low number of contigs, particularly compared to those for other reef fishes with scaffold-level genomes based on Illumina short-read approaches. The *C. cyanea* genome contigs closely mapped against chromosome-scale genomes from other Pomacentrids, with several homologous contig-chromosome pairs (Figure 4, Table 4). The high quality of the genome assembled here is indicative that the sequencing depth, choice of technology, and downstream pipelines were sufficient to retrieve long contigs and achieve a near chromosome-scale genome assembly.

Several architectural variations in the genome of *C. cyanea* were detected by comparison with the five reference genomes. Some contigs showed homology with their reference chromosomes (Figure 4, Table 4, Supplementary Tables 3, 4, and 5 in Figshare [60]). However, others showed rearrangements that ranged from inversions with or without frame shifts to a possible rearrangement of ancestral chromosomes in the Cheiloprionini lineage (Figure 4B). In particular, over 50% of the bases of contig-1 were

aligned to at least two chromosomes of all reference genomes, with about half of the bases mapping to one chromosome and half to another (Figure 4, Supplementary Table 5 in Figshare [60]). No *C. cyanea* contigs robustly scaffolded to chromosome 24 of *Ac. polyacanthus* and *D. trimaculatus* (Supplementary Table 3 in Figshare [60]). As chromosome 24 of *D. trimaculatus* was reported to be highly rearranged [3], the absence of alignment with this chromosome highlights that the rearrangement occurred in the Chrominae branch, which separated from the Pomacentrinae branch, to which *Amphiprion Acanthochromis*, and *Chrysiptera* belong, over 50 million years ago [64, 65]. The relative distance of *Chrysiptera* to *D. trimaculatus* compared with the other reference species is also illustrated by contig-18, which is the contig showing the highest rate of alignment with most genomes. Indeed, 67.7–75.1% of the bases of contig-18 were aligned to the *A. clarkii, A. ocellaris, A. percula,* and *Ac. polyacanthus* reference genomes but only 34% with *D. trimaculatus* (Figure 4A). Lastly, the similarity in results when comparing the *C. cyanea* assembly with each of the four *Amphiprion* and *Acanthochromis* reference genomes can also be linked to the evolutive history of Pomacentrids. *C. cyanea* is part of the Cheiloprionini, which split from the other Pomacentrinae, notably from the branch to which *Amphiprion* and *Acanthochromis* belong, over 35 million years ago, leading to similar differences in the comparisons to all of these genomes [64, 65].

## CONCLUSION

In this study, the first genome assembly for a *Chrysiptera* species was generated with 91 contigs. The contigs were successfully aligned to related reference genomes, allowing for forays into the genomic architecture of Pomacentrids with respect to their evolutive relationship. *C. cyanea* is easy to breed and maintain in aquaria and is highly abundant in Indo-Pacific coastal waters, particularly in Okinawa, Japan. The generation of this high-quality genome will further the potential of this species as a coral reef model species for research questions requiring biomolecular approaches.

## DATA AVAILABILITY

The genome and transcriptome sequencing reads are deposited in the NCBI GenBank database under the BioProject PRJNA1167451. The genome assembly, annotation, and proteome for *C. cyanea,* as well as supplemental tables, BUSCO outputs and detailed scripts for each genome assembly and annotation step, are available in FigShare [60].

## ABBREVIATIONS

COI, cytochrome *c* oxidase subunit I; IPA, Improved Phased Assembler; TL, total length; TPM, transcripts per million.

## DECLARATIONS

### Ethics approval and consent to participate

The experimental procedure was conducted in accordance with ethical procedures recommended by the Okinawa Institute of Science and Technology Graduate University.

### Competing interests

The authors declare no competing interests.

## Authors' contributions

EG: Investigation, Formal Analysis, Funding Acquisition, Writing, SM: Project Administration, Investigation, HT: Investigation, MH: Supervision, Formal Analysis, Software, Writing, VL: Conceptualization, Supervision, Project Administration, Funding Acquisition, Writing.

## Funding

Funding for this research was provided by the Okinawa Institute of Science and Technology Graduate University SHINKA Grant as well as the Iwatani Naoji Memorial Foundation through the 49th Iwatani Science and Technology Research Grant FY2023.

## Acknowledgements

The authors would like to thank the OIST Sequencing Section for the library preparation and sequencing, the Scientific Computing and Data Analysis section of Core Facilities at OIST, as well as Stefano Vianello for providing the drawings of *Chrysiptera cyanea*.

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
