## [Editor Report]

Editor’s AssessmentAmong hot topics in coral reef research, the difference between anemonefish and other damselfish is currently a popular area of research. In this study the authors provide a new high-quality non-anemonefish genome, which will be of high relevance to further the depth of such analyses. In this case of the sapphire damselfish Chrysiptera cyanea, a widely distributed damselfish in the Indo-Pacific area, often studied to elucidate the roles of various environmental controls on their reproduction, and investigate related hormonal processes To further the potential of biomolecular analyses based on this species, this study generated the first genome of a Chrysiptera fish from a male individual collected in Okinawa, Japan. Using PacBio and HiFI long-read sequencing with 94.5x coverage, a chromosome-scale genome was assembled and 28,173 genes identified and annotated. Peer review gathered more parameters and details on the quality, and the final assembly comprised of 896 Mb pairs across 91 contigs, and a BUSCO completeness of 97.6%. This reference genome should therefore be of high value for future genetic-based approaches, from population structure to gene expression analyses.Editor’s AssessmentAmong hot topics in coral reef research, the difference between anemonefish and other damselfish is currently a popular area of research. In this study the authors provide a new high-quality non-anemonefish genome, which will be of high relevance to further the depth of such analyses. In this case of the sapphire damselfish Chrysiptera cyanea, a widely distributed damselfish in the Indo-Pacific area, often studied to elucidate the roles of various environmental controls on their reproduction, and investigate related hormonal processes To further the potential of biomolecular analyses based on this species, this study generated the first genome of a Chrysiptera fish from a male individual collected in Okinawa, Japan. Using PacBio and HiFI long-read sequencing with 94.5x coverage, a chromosome-scale genome was assembled and 28,173 genes identified and annotated. Peer review gathered more parameters and details on the quality, and the final assembly comprised of 896 Mb pairs across 91 contigs, and a BUSCO completeness of 97.6%. This reference genome should therefore be of high value for future genetic-based approaches, from population structure to gene expression analyses.

---

## [Reviewer Report]

Reviewer name and names of any other individual's who aided in reviewer Yue SongDo you understand and agree to our policy of having open and named reviews, and having your review included with the published papers. (If no, please inform the editor that you cannot review this manuscript.)YesIs the language of sufficient quality?YesPlease add additional comments on language quality to clarify if needed
Are all data available and do they match the descriptions in the paper? NoAdditional CommentsThe authors have provided clues for accessing the data in public databases such as NCBI, but it seems that the data has not been released; At least, I haven't been able to obtain available data using the provided accession number (e.g. PRJNA1167451). I'm not sure if I've missed any information, but I believe it would be better if the data could be easily accessible to the public.Are the data and metadata consistent with relevant minimum information or reporting standards? See GigaDB checklists for examples <a href="http://gigadb.org/site/guide" target="_blank">http://gigadb.org/site/guide</a>YesAdditional CommentsIs the data acquisition clear, complete and methodologically sound?NoAdditional CommentsThe authors used PacBio's third-generation sequencing technology for genome sequencing, which has become a "necessary option" for obtaining high-quality genomes in current genomic research. However, they did not further advance on the path of "assembling a chromosome-level genome" based on this version. Providing a chromosome-level genome would likely be more meaningful.Is there sufficient detail in the methods and data-processing steps to allow reproduction?NoAdditional CommentsRegarding the genome assembly and annotation process, the method described by the authors is overly simplistic and lacks detailed information on the parameters and procedures used. This makes it difficult for other researchers to effectively replicate the results described in the article.Is there sufficient data validation and statistical analyses of data quality? NoAdditional CommentsThe authors have calculated the N50 of contigs and the completeness of BUSCO genes, which are indeed two commonly used indicators for assessing the quality of genome assemblies. However, it is still challenging to gain a clear understanding of the assembly quality based solely on these two indicators. Could other measurements be added, such as comparing the continuity and completeness of the assembly with those of closely related species or other comparable species' genomes? Additionally, there is a point that is difficult to understand: the authors report a BUSCO completeness of approximately 94% for the genome, yet a BUSCO completeness of 97% for the gene set. It is puzzling how BUSCO genes that are not annotated in the genome can still be present in the gene set.Is the validation suitable for this type of data?YesAdditional CommentsIs there sufficient information for others to reuse this dataset or integrate it with other data?NoAdditional CommentsAs I mentioned earlier, the authors did not provide detailed information about the processing procedures and parameters, which makes it difficult for other researchers to replicate their results.Any Additional Overall Comments to the AuthorIt is recommended that the authors provide a detailed description of the methods and easily accessible data retrieval methods. It would be even better if the authors could further provide a chromosome-level genome, as T2T (telomere-to-telomere) level genomes are becoming increasingly popular.RecommendationMinor Revision

---

## [Reviewer Report]

Reviewer name and names of any other individual's who aided in reviewer Darrin T. SchultzDo you understand and agree to our policy of having open and named reviews, and having your review included with the published papers. (If no, please inform the editor that you cannot review this manuscript.)YesIs the language of sufficient quality?YesPlease add additional comments on language quality to clarify if needed
Are all data available and do they match the descriptions in the paper? YesAdditional CommentsThe genome is also not yet on NCBI, but it would be good to upload it.Are the data and metadata consistent with relevant minimum information or reporting standards? See GigaDB checklists for examples <a href="http://gigadb.org/site/guide" target="_blank">http://gigadb.org/site/guide</a>YesAdditional CommentsI suggest later that there should be more information about the HiFi library preparation details, as the manuscript lacks them and it appears to be a non-standard (large insert size) library.Is the data acquisition clear, complete and methodologically sound?NoAdditional CommentsSee above comment-Is there sufficient detail in the methods and data-processing steps to allow reproduction?NoAdditional CommentsNo parameters are provided for the genome assembly software, for read trimming, or for other software used.Is there sufficient data validation and statistical analyses of data quality? NoAdditional CommentsSee extended comments - the read data could use more QC, as well as the genome assembly.Is the validation suitable for this type of data?NoAdditional CommentsIs there sufficient information for others to reuse this dataset or integrate it with other data?YesAdditional CommentsThere is a degree of information missing about the data, but another researcher could use them for their study.Any Additional Overall Comments to the AuthorThank you for the opportunity to review the work, The genome of the sapphire damselfish Chrysiptera cyanea: a new resource to support further investigation of the evolution of Pomacentrids, by Gairin and colleagues. In this manuscript, the authors collect an individual of the pomocentrid fish, Chrysiptera cyanea, in Okinawa, Japan. After isolating DNA, the sequencing center at OIST prepared and sequenced a SMRT sequencing library. Additionally, the authors generated some bulk RNA-seq data and sequenced it on the Illumina platform. The authors assembled the genome with two assemblers, and performed some comparisons of the C. cyanea contigs aligned to the chromosome-scale scaffolds of closely related pomacentrids. Given my background, I will mostly comment on the genomic analyses. I appreciate the authors' diligence in exploring different genome assembly methods and their efforts in running BUSCO and QUAST to QC the assemblies. The DNA sequencing data and assembly produced contigs that align well with the chromosomes of closely related species (which is convenient for comparative genomics!), and the manuscript presents a solid foundation for better understanding the chromosomal evolutionary history of the Pomacentridae. While this work represents an important step toward providing a new genomic resource for Chrysiptera cyanea, I see a few areas where the manuscript could be refined to enhance it as a community resource: (1) More information about data generation: Including additional details about the HiFi library preparation, specifically the chemistries used, the number of SMRT cells sequenced, and the bioinformatics steps used to generate the HiFi reads, would improve the manuscript's clarity and reproducibility. I have some questions regarding whether these libraries were prepared for HiFi sequencing: the reported mean read length of 25kbp is 10kbp longer than the standard HiFi library insert size; and the reported amount of bases in the reads, 84 Gbp, is more data than one would expect from a single CCS-processed SMRT cell, but could be the amount of data produced from one CLR run. Characterizing the quality score vs read length distribution could be helpful to characterize the read data. Clarifying these steps taken before the genome was assembled would strengthen the reliability of these reads as a resource. (2) Incorporating a few more important quality control (QC) steps would better clarify the completeness of the genome assembly. For instance, an estimate of genome size from the HiFi reads could be performed with jellyfish and GenomeScope, taking advantage of the k-mer fidelity of HiFi reads. This would provide a more conclusive estimate than the current comparison. Additionally, steps such as checking for contamination and providing an explanation for decisions like haplotig removal would make the assembly process more transparent. Lastly, supplementing the QC analysis with Merqury will provide a reliable answer to how complete the assembly represents the information in the individual HiFi reads in a way that complements BUSCO and QUAST. (3) The initial analyses of chromosome structure are a promising look into some yet-unexplored chromosomal changes in the Pomacentridae, and I think that incorporating a deeper phylogenetic analysis would build on this strength. Situating the chromosomal findings within a phylogenetic framework could provide stronger support, or actually resolve, the evolutionary interpretations presented. Doing this analysis likely could also help resolve whether the structures seen are genome misassemblies, or instead reflect lineage-specific chromosomal changes. The authors could supplement their beautiful figures using other tools that leverage whole-genome alignments and chromosome visualization to help answer these questions. One tool to try for two-genome comparisons, that the authors may have explored already in place of their ggplot script, is D-GENIES. Overall, this is a valuable resource, and I commend the authors for taking the steps to analyze the chromosomal evolutionary history within the pomacentrids. I look forward to seeing the authors’ future contributions to the field of genomics and chromosome evolution. Minor Points Line 125: Sharing the specific Trimmomatic settings used would enhance the reproducibility of the RNA-seq data processing. The parameters for genome assembly should also be added. Line 212: Are there any replicates for the RNA-seq data? Line 294: Consider uploading the assembly to NCBI for broader visibility and accessibility.RecommendationMinor Revision